# Synthesis, Characterization, and Biological Properties of the Copper(II) Complexes with Novel Ligand: *N*-[4-({2-[1-(pyridin-2-yl)ethylidene]hydrazinecarbothioyl}amino)phenyl]acetamide

Roman Rusnac [1] , Olga Garbuz [1,2,*] , Yurii Chumakov [3] , Victor Tsapkov [1] , Christelle Hureau [4] , Dorin Istrati [5] and Aurelian Gulea [2,*]

1 Laboratory of Advanced Materials in Biopharmaceutics and Technics, Moldova State University, 60 Mateevici Street, MD-2009 Chisinau, Moldova

2 Biological Invasions Research Center, Institute of Zoology, Moldova State University, 1 Academiei Street, MD-2028 Chisinau, Moldova

3 Laboratory of Physical Methods of Solid State Investigation "Tadeusz Malinowski", Institute of Applied Physics, MD-2028 Chisinau, Moldova

4 Laboratoire de Chimie de Coordination UPR 8241, Centre National de la Recherche Scientifique, 31400 Toulouse, France

5 Department of Dentistry, University of Medicine and Pharmacy "Nicolae Testemitanu", 165 Stefan cel Mare si Sfant Bd., MD-2004 Chisinau, Moldova

* Correspondence: olga.garbuz@sti.usm.md (O.G.); guleaaurelian@gmail.com (A.G.)

**Abstract:** For the first time, a thiosemicarbazone-type ligand containing a paracetamol structural unit was synthesized. Five new coordination compounds based on copper(II) salts: $[Cu(L)CH_3COO]$ (**1**), $[\{Cu(L)Cl\}_2]\cdot H_2O$ (**2**), $[Cu(L)H_2O\cdot DMF]NO_3$ (**3**), $[Cu(L)Br]$ (**4**), $[Cu(L)H_2O]ClO_4$ (**5**), were obtained, where **HL** is *N*-[4-({2-[1-(pyridin-2-yl)ethylidene]hydrazinecarbothioyl}amino)phenyl]acetamide. The new **HL** was characterized by NMR, FTIR, spectroscopy, and X-ray crystallography. All copper(II) coordination compounds were characterized by elemental analysis, FTIR, EPR spectroscopy, and molar electrical conductivity. Furthermore, single crystal X-ray diffraction analysis elucidated the structures of thiosemicarbazone **HL** as well as complexes **1**–**3**. All compounds were tested for antimicrobial, antifungal, and antioxidant activities, and their toxicity to *Daphnia magna* was studied. Biological evaluation has revealed that most of the synthesized compounds demonstrate promising antibacterial, antifungal, and antioxidant activities. In many cases, their antibacterial/antifungal activity is comparable to that of certain drugs used in medicine for these purposes, and in some cases, even surpasses them. **HL** and complexes **2**–**5** exhibit antioxidant activity that surpasses that of Trolox. Furthermore, **HL** and complex **2** display virtually no toxicity to *D. magna*.

**Keywords:** coordination compound; thiosemicarbazone; crystal structure; antioxidant activity; antimicrobial activity; antifungal activity; toxicity

## 1. Introduction

The study of new materials in the field of coordination chemistry is increasing day by day, leading to the discovery of substances with advanced biological properties compared to the drugs currently used in medicinal practice. A class of organic compounds called thiosemicarbazones is expected to be the most promising due to a wide range of biological activities, including antituberculosis [1–4], antineoplastic activity [2], anticancer [5], antioxidant [6], antiviral [7], antimicrobial [8,9], antifungal [1,10–12], anticonvulsant [13–15], antiproliferative activity [16], anticancer activity [17]. Interest in this family of compounds has grown significantly over the years, from the first report in 1940 to the thousands of papers published by 2022.

Coordination compounds based on 3*d* metal ions have d orbitals partially coupled with electrons and exhibit different variable oxidation states, playing an important role in redox processes. They hold promise in the development of pharmaceutical agents [18].

The identification of potential new antibactericidal/antifungicidal drugs with increased efficiency based on coordinative combinations is in vogue, their stake would be low side effects; and overcoming the resistance achieved by current drugs [19–24].

Thiosemicarbazones have attracted the attention of researchers [18,25] for four key aspects in the field of coordination chemistry: high coordination tendency; formation of coordinative complexes with increased stability; high selectivity capacity; and ability to form macrocycles.

Thiosemicarbazones possess a dual aspect based on biological [26,27] and computational principles [28]. From a biological standpoint, thiosemicarbazones exhibit various pharmacological properties. From a theoretical aspect, they are ideal targets for computational studies due to their donor-acceptor capabilities in the development of therapeutic agents [29].

One important biological property of thiosemicarbazones is their ability to inhibit ribonucleotide reductase (RR) synthesis [30]. Substitution in the *para* position is a decisive factor in the antifungal potential of 2-acetylpyridine thiosemicarbazones [31].

The strategy of choosing the thiosemicarbazone skeleton was based on two positions, namely: position one—the thiosemicarbazone fragment must contain a fragment that is easily metabolized by the human body and does not show toxic effects. The second position is that the part responsible for the selective biological effect (pyridin-2-yl) is included in the thiosemicarbazone phial casing. Thus, the hypothesis from those elucidated could be reproduced in the figure below (Figure 1):

**Figure 1.** Structural thiosemicarbazone formula of **HL**.

The aim of the present investigation is the synthesis, characterization, and study of antibacterial, antifungal, and antioxidant activities of Paracetamol (4-Aminoacetanilide) with a thiosemicarbazone fragment (**HL**) and Cu(II) coordination compounds with **HL**: [Cu(L)CH$_3$COO] (**1**), [{Cu(L)Cl}$_2$]·H$_2$O (**2**), [Cu(L)H$_2$O·DMF]NO$_3$ (**3**), [Cu(L)Br] (**4**), [Cu(L)H$_2$O]ClO$_4$ (**5**).

## 2. Results and Discussion

In this work, a new thiosemicarbazone based on 4-aminoacetanilide was synthesized, functionalized according to the organic synthesis procedure: synthesis of the isothiocyanate group and hydrazone following the nucleophilic addition reaction with the formation of the **HL** ligand (Figure 2).

**Figure 2.** Scheme of synthesis of thiosemicarbazone **HL**.

The thiosemicarbazone **HL** was characterized by FT-IR, ¹H NMR, and ¹³C NMR spectroscopy. Its structure was determined using X-ray diffraction analysis. Complexes **1–5** were synthesized by the interaction of ethanolic solution of *N*-[4-({2-[1-(pyridin-2-yl)ethylidene]hydrazinecarbothioyl}amino)phenyl]acetamide and copper(II) salts (**1–5**) in a 1:1 molar ratio. The composition of the thiosemicarbazone **HL** and complexes **1–5** was confirmed using elemental analysis data.

In the NMR spectra (Figures S1 and S2), two tautomeric forms (Figure 3) were determined in the case of thiosemicarbazone **HL**. The thiol tautomeric form is identified by the characteristic peak of the SH group at the chemical shift of 14.54 ppm in the hydrogen spectrum (¹H-NMR) corresponding to data from the specialized literature [32,33]. The ionic tautomeric form of the carbon spectrum (¹³C-NMR) is found at the chemical shift 177.82 ppm, which is consistent with the literature data [16,34,35].

**Figure 3.** Tautomeric forms of **HL**.

## 2.1. Structural Characterization of Ligand HL and Coordination Compounds 1–3

The X-ray structures of **HL** and coordination compounds **1–3** are presented in Figures 4 and 5, and Tables 1–3. In the molecule of **HL** (Figure 4a) the substituents at the N(2)–C(1) bond are in the E position. The A(S(1)-N(1)-N(2)-N(4)-C(1)-C(2)) core is practically planar within 0.1 Å and the dihedral angle between the given core and pyridine ring is equal to 10.4°. However, the **HL** is nonplanar because the dihedral angle between the best plane of A and the benzene ring is 58.2°. In the crystal, the ligand **HL** forms the dimers via N(2)-H... S(1) hydrogen bonds (HB), which are liked by N(5)-H... O(2) HB into the layers along the a-axis (Table 3, Figure 5a).

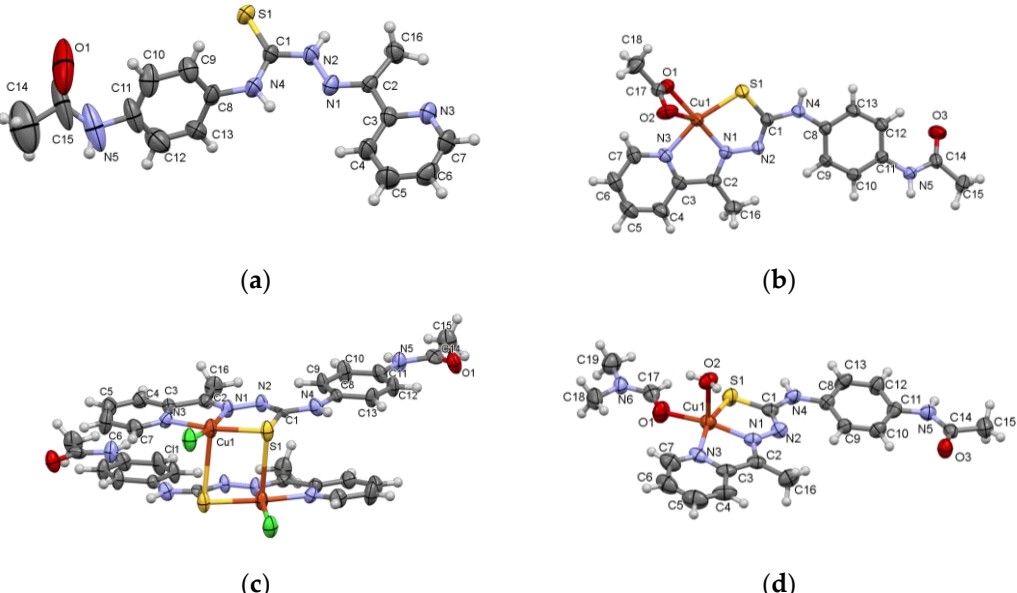

**Figure 4.** The molecular structures and atom-numbering of **HL** (**a**), [Cu(L)CH₃COO] in **1** (**b**), [{Cu(L)Cl}₂]·H₂O in **2** (**c**), [Cu(L)H₂O·DMF]NO₃ in **3** (**d**).

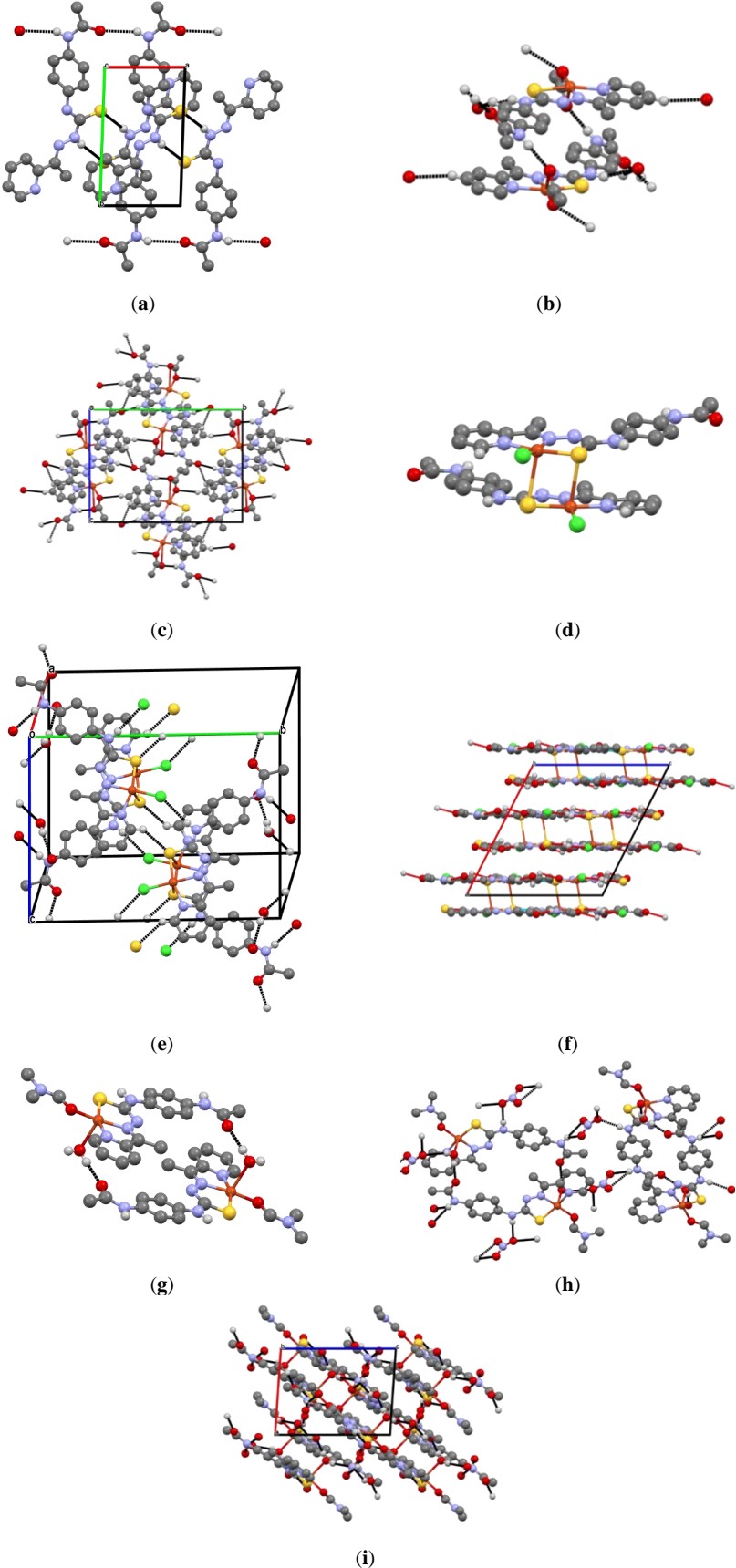

**Figure 5.** The crystal packing fragments of (**a**) **HL**, (**c**) **1**, (**f**) **2**, (**i**) **3**, dimers in (**b**) **1**, (**d**) **2**, (**g**) **3**, layers in (**e**) **2**, and (**h**) **3**.

**Table 1.** Crystal and structure refinement data for **HL** and **1–3**.

| Identification Code | HL | [Cu(L)CH$_3$COO] (1) |
|---|---|---|
| CCDC | 2213214 | 2213213 |
| Empirical formula | C$_{16}$H$_{17}$N$_5$OS | C$_{18}$H$_{19}$N$_5$O$_3$SCu |
| Formula weight | 327.40 | 448.98 |
| Temperature/K | 293(2) | 293(2) |
| Crystal system | triclinic | monoclinic |
| Space group | P-1 | P2$_1$/c |
| a/Å | 5.6303(7) | 8.5347(4) |
| b/Å | 10.2646(10) | 17.6158(5) |
| c/Å | 14.941(2) | 13.5270(5) |
| α/° | 108.622(11) | 90 |
| β/° | 92.330(11) | 105.779(5) |
| γ/° | 91.118(9) | 90 |
| Volume/Å$^3$ | 817.11(18) | 1957.08(13) |
| Z | 2 | 4 |
| ρ$_{calc}$g/cm$^3$ | 1.331 | 1.524 |
| μ/mm$^{-1}$ | 0.210 | 1.252 |
| F(000) | 344.0 | 924.0 |
| Crystal size/mm$^3$ | 0.55 × 0.7 × 0.06 | 0.21 × 0.18 × 0.15 |
| Radiation | MoKα (λ = 0.71073) | MoKα (λ = 0.71073) |
| 2Θ range for data collection/° | 5.94 to 50.076 | 6.26 to 50.084 |
| Index ranges | −6 ≤ h ≤ 5, −12 ≤ k ≤ 9, 17 ≤ l ≤ 17 | −10 ≤ h ≤ 10, −20 ≤ k ≤ 15, −16 ≤ l ≤ 11 |
| Reflections collected | 5061 | 7038 |
| Independent reflections | 2881 [R$_{int}$ = 0.0413, R$_{sigma}$ = 0.1093] | 3429 [R$_{int}$ = 0.0363, R$_{sigma}$ = 0.0613] |
| Data/restraints/parameters | 2881/6/213 | 3429/0/256 |
| Goodness-of-fit on F$^2$ | 1.035 | 1.013 |
| Final R indexes [I ≥2σ (I)] | R$_1$ = 0.0821, wR$_2$ = 0.1964 | R$_1$ = 0.0445, wR$_2$ = 0.0872 |
| Final R indexes [all data] | R$_1$ = 0.1530, wR$_2$ = 0.2358 | R$_1$ = 0.0709, wR$_2$ = 0.0975 |
| Largest diff. peak/hole/e Å$^{-3}$ | 0.62/−0.37 | 0.33/−0.30 |

| Identification code | [{Cu(L)Cl}$_2$]·H$_2$O (2) | [Cu(L)H$_2$O·DMF]NO$_3$ (3) |
|---|---|---|
| CCDC | 2213215 | 2213216 |
| Empirical formula | C$_{16}$H$_{18}$N$_5$O$_2$SClCu | C$_{19}$H$_{25}$N$_7$O$_6$SCu |
| Formula weight | 443.40 | 543.07 |
| Temperature/K | 293(2) | N/A |
| Crystal system | monoclinic | monoclinic |
| Space group | C2/c | P2$_1$/n |
| a/Å | 15.166(3) | 8.1545(7) |
| b/Å | 18.8615(15) | 26.686(2) |
| c/Å | 14.090(2) | 10.8802(9) |
| α/° | 90 | 90 |
| β/° | 117.14(2) | 93.856(8) |
| γ/° | 90 | 90 |
| Volume/Å$^3$ | 3586.6(12) | 2362.3(3) |
| Z | 8 | 4 |
| ρ$_{calc}$g/cm$^3$ | 1.642 | 1.5268 |
| μ/mm$^{-1}$ | 1.505 | 1.063 |
| F(000) | 1816.0 | 1126.3 |
| Crystal size/mm$^3$ | 0.32 × 0.06 × 0.01 | 0.43 × 0.18 × 0.05 |
| Radiation | MoKα (λ = 0.71073) | Mo Kα (λ = 0.71073) |
| 2Θ range for data collection/° | 6.174 to 50.5 | 5.92 to 50.5 |
| Index ranges | −18 ≤ h ≤ 11, −22 ≤ k ≤ 14, −9 ≤ l ≤ 16 | −11 ≤ h ≤ 8, −37 ≤ k ≤ 37, −15 ≤ l ≤ 15 |
| Reflections collected | 3484 | 9282 |
| Independent reflections | 2313 [R$_{int}$ = 0.0852, R$_{sigma}$ = 0.1628] | 4091 [R$_{int}$ = 0.0634, R$_{sigma}$ = 0.1693] |
| Data/restraints/parameters | 2313/0/240 | 4091/0/314 |
| Goodness-of-fit on F$^2$ | 0.892 | 1.011 |
| Final R indexes [I ≥ 2σ (I)] | R$_1$ = 0.0713, wR$_2$ = 0.1092 | R$_1$ = 0.0683, wR$_2$ = 0.1090 |
| Final R indexes [all data] | R$_1$ = 0.1564, wR$_2$ = 0.1432 | R$_1$ = 0.1396, wR$_2$ = 0.1442 |
| Largest diff. peak/hole/e Å$^{-3}$ | 0.45/−0.40 | 1.37/−0.82 |

**Table 2.** Selected bond lengths (Å) and angles (deg) in fragments of thiosemicarbazones in **HL** and **1–3**.

| Bond | d, Å | | | |
|---|---|---|---|---|
| | **HL**[1] | **1** | **2** | **3** |
| Cu1-S1 | | 2.2452(11) | 2.257(3) | 2.2458(18) |
| Cu1-O1(Cl1) | | 1.942(3) | 2.273(3) | 1.986(4) |
| Cu1-N1 | | 1.948(3) | 2.005(8) | 1.972(4) |
| Cu1-N3 | | 2.014(3) | 1.981(8) | 2.013(5) |
| Cu1-O2(S1′) | | 2.755(3) | 2.982(3) | 2.338(4) |
| S1-C1 | 1.672(5) | 1.747(3) | 1.757(8) | 1.744(6) |
| N1-N2 | 1.367(5) | 1.372(4) | 1.370(11) | 1.376(7) |
| N1-C2 | 1.288(7) | 1.301(4) | 1.286(13) | 1.288(9) |
| N2-C1 | 1.354(7) | 1.322(4) | 1.301(11) | 1.311(7) |
| N3-C3 | 1.339(7) | 1.362(4) | 1.343(12) | 1.369(8) |
| N4-C1 | 1.339(7) | 1.343(4) | 1.378(12) | 1.351(7) |
| C2-C3 | 1.475(6) | 1.462(5) | 1.487(15) | 1.466(11) |
| **Angle** | **ω°** | | | |
| S1-Cu1-O1(Cl1) | | 97.24(7) | 98.16(10) | 98.65(13) |
| S1-Cu1-N1 | | 84.71(8) | 83.4(2) | 84.43(15) |
| S1-Cu1-N3 | | 164.46(8) | 163.2(2) | 162.20(16) |
| S1-Cu1-O2(S1′) | | 100.77(6) | 96.07(11) | 103.02(13) |
| O1(Cl1)-Cu1-N1 | | 174.04(11) | 168.1(3) | 170.24(18) |
| O1(Cl1)-Cu1-N3 | | 97.82(10) | 98.2(2) | 94.5(2) |
| N1-Cu1-N3 | | 80.68(11) | 79.8(3) | 80.7(2) |
| O1(Cl1)-Cu1-O2(S1′) | | 52.17(10) | 91.31(10) | 88.55(16) |
| N1-Cu1- O2(S1′) | | 133.09(10) | 100.3(3) | 99.82(16) |
| N3-Cu1- O2(S1′) | | 85.55(9) | 87.3(3) | 89.22(19) |
| Cu1-S1-C1 | | 94.78(11) | 95.5(3) | 94.86(19) |
| Cu1-N3-C3 | | 112.2(2) | 114.5(7) | 112.3(5) |
| Cu1-N1-N2 | | 123.58(19) | 124.1(6) | 122.9(3) |
| Cu1-N1-C2 | | 118.1(2) | 117.6(7) | 117.3(4) |
| N2-N1-C2 | 118.3(4) | 118.3(3) | 118.3(8) | 119.8(5) |
| N1-N2-C1 | 119.1(4) | 111.2(2) | 111.6(6) | 111.3(4) |
| S1-C1-N2 | 119.9(3) | 125.4(2) | 125.3(7) | 126.0(4) |
| N1-C2-C3 | 115.1(4) | 113.5(3) | 114.1(8) | 114.5(6) |
| N3-C3-C2 | 116.5(4) | 115.4(3) | 114.0(9) | 115.0(6) |

**Table 3.** Hydrogen bond distances (Å) and angles (deg) in **HL** and **1–3**.

| D–H···A | d(H···A) | d(D···A) | ∠(DHA) | Symmetry Transformation for Acceptor |
|---|---|---|---|---|
| **HL** | | | | |
| N2-H... S1 | 2.86 | 3.714(4) | 174.0 | 2-x,1-y,2-z |
| N5-H... O2 | 1.85 | 2.63(2) | 150.0 | x,-1 + y,z |
| **1** | | | | |
| N4-H... O3 | 2.14 | 2.937(4) | 153.0 | x,3/2-y,-1/2 + z |
| N5-H... O2 | 2.0 | 2.854(4 | 177.0 | 1-x,1-y,-z |
| C5-H... O1 | 2.57 | 3.477(4) | 164.0 | x,3/2-y,-1/2 + z |
| C13-H... O3 | 2.54 | 3.211(4) | 130.0 | 2-x,-1/2 + y,-1/2-z |
| **2** | | | | |
| O2-H... O1 | 2.15 | 2.878(11) | 144.0 | x. y, z |
| N4-H... Cl1 | 2.81 | 3.666(8) | 171.0 | x,2-y,1/2 + z |
| N5-H... O2 | 2.26 | 3.105(13) | 166/0 | x,1-y,-1/2 + z |
| C7-H... S1 | 2.72 | 3.459(10) | 137/0 | x,2-y,-1/2 + z |
| **3** | | | | |
| O2-H... O3 | 1.804 | 2.678(6) | 174.0 | 2-x,1-y,2-z |
| O2-H... O5 | 2.33 | 3.088(7) | 165.0 | x, y, z |
| O2-H... S1 | 2.86 | 3.214(5) | 110.0 | 1-x,1-y,1-z |
| N4-H... O5 | 2.13 | 2.985(7) | 174.0 | 1-x,1-y,1-z |
| N5-H... O4 | 2.28 | 3.133(9) | 170.0 | 3/2-x,-1/2 + y,3/2-z |
| N5-H... O6 | 2.52 | 3.227(9) | 140.5 | 3/2-x,-1/2 + y,3/2-z |

In complexes **1–3**, the ligand **HL** acts as a mononegative tridentate around the metallic ions through the SNN set of donor atoms. The studied complexes are five-coordinated in a distorted square–pyramidal coordination geometry. Deprotonation of N(2) atom in **1–3** has led to the decrease of N(2)-C(1) bond distance if compared with that in **HL**. The bond lengths S(1)-C(1) in these complexes are increased due to the coordination of sulfur atoms to central metals and the maximal changing of this bond is observed in **2** which is equal to 1.757(8) Å (Table 2). However, the composition of the coordination polyhedron of the central atom in these compounds is different. Its basal plane includes three donor atoms of the **L**, oxygen atoms O(1) of coordinated $CH_3COO$ and DMF molecules in **1, 3**, and chlorine ion in **2** (Figure 4b–d, Table 2). The deviations of these atoms from their mean plane are within 0.1 Å, while the Cu atoms deviate from these planes by 0.01, 0.13, and 0.17 Å toward the apexes of the pyramids. The apexes of the metal's coordination pyramids in **1**, and **3** are occupied by oxygen atoms O(2) of $CH_3COO$ and water molecules with distances of 2.755 (3) and 2.338 (4) Å, respectively. In **2**, the complexes form the centrosymmetric dimers, and the apex of the coordination pyramid is occupied by the sulfur atom of the adjacent complex with a distance of 2.982(3) Å (Figure 5d). In the crystal of **1**, the complexes are joined by N5-H... O2 hydrogen bonds in centrosymmetric dimers, which are further linked by C5-H... O1, N4-H... O3, and C13-H... O3 HB, forming the 3D hydrogen bonding networks (Figure 5b,c). In **2**, dimers linked by water molecules as well as N(4)-H... Cl(1) and C(7)-H... S1 HB form the layers along *a-* axis (Figure 5d–f). The water molecules in crystal **3** form the centrosymmetric dimers, which are joined by the $NO_3$-group in the layers along the same direction (Figure 5g–i, Table 3). In both **2** and **3**, van der Waals interaction occurs between the layers.

Molar conductivity was determined in an ethanol-water (3:1) solution in the range of 36–106 μS/cm with the ratio of electrolyte 1:1 or non-electrolyte being determined. Coordinative combinations with anions $CH_3COO^-$, $Cl^-$, $NO_3^-$, $Br^-$, $ClO_4^-$ form the internal or external sphere, which upon solvolysis can be in equilibrium with the undissociated form. The activity of molar conductivity date in case coordination combinations **1**, **2**, and **4** are non-electrolytes. For compounds **3** and **5**, the acidic residue is found in the outer sphere.

In the IR spectrum (Figure S3) of thiosemicarbazone HL, the presence of groups was determined to be: C=S (thionic), C=N(azomethenic), 1,4-disubstituted Aryl, N-H(hydrazinic). The coordination combinations show bands in the IR spectra characteristic of the coordination of nitrogen, sulfur, and pyridinic nitrogen donor atoms with the central copper(II) atom. In the IR spectrum of complex 1, the acetate ion is coordinated bidentate, which generates two bands with high intensity at 1603–1563 $cm^{-1}$ characteristic $\nu_a(COO)$ and $\nu_s(COO)$. In the case of the nitrate ion from complex 3 in the IR spectrum, a band at 1309 and 1251 $cm^{-1}$ characteristic of ionic form is presented (Figures S3–S8) [36].

The shift of the bands from higher to lower wave numbers in the case of the azomethine group from the free ligand 1611 $cm^{-1}$ to 1584–1531 $cm^{-1}$ in the case of coordination combinations **1–5** indicates the coordination at the central atom.

The EPR signature of complexes **1–5** were measured in DMSO at 200 μM (Figure S9), complexes **1** and **3–5** show similar signatures while that of **5** is the one which is the best resolved while an additional broad signal centered near g=2 contributes to the overall spectra of **1**, **3** and **4**. Two different fingerprints corresponding to two mononuclear Cu(II) species (called I and II) can be observed and distinguished by their EPR parameters. Species I has the following parameters: $A_{//}$ = 180 G, $g_{//}$ = 2.20, and species II has: $A_{//}$ = 170 G, $g_{//}$ = 2.15. The ratio between these two species depends on the complex under study. This may indicate that these four complexes evolve and form the same two mononuclear EPR-sensitive species in solution but in different proportions. Complex **2** exhibits a different fingerprint with an ill-resolved pattern in line with the binuclear species mainly kept in solution.

## 2.2. Biological Activity

### 2.2.1. Antimicrobial and Antifungal Activity

The literature indicates that thiosemicarbazones and their copper(II) complexes often demonstrate antimicrobial and antifungal properties [10]. Therefore, the antimicrobial and antifungal characteristics of the synthesized compounds were examined. The ligand **HL** and its complexes **1–5** were assessed against Gram-positive bacteria (*Staphylococcus aureus* ATCC 25923), Gram-negative bacteria (*Escherichia coli* ATCC 25922), and fungal strains (*Candida albicans* ATCC 10231). The MIC (minimum inhibitory concentration, $\mu g \, mL^{-1}$), MBC (minimum bactericidal concentrations, $\mu g \, mL^{-1}$), and MFC (minimum fungicidal concentrations, $\mu g \, mL^{-1}$) values for the compounds against bacteria and fungi are shown in Table 4. Nitrofurazone [37] and miconazole [38] were used as standard drugs for comparing antimicrobial and antifungal activities, respectively.

**Table 4.** Minimal inhibitory, and bactericidal/fungicidal concentrations ($\mu g \, mL^{-1}$) of **HL,** and copper(II) complexes **1–5** in relation to test microbes and fungi.

| Compound | *Staphylococcus aureus* (ATCC 25923) | | *Escherichia coli* (ATCC 25922) | | *Candida albicans* (ATCC 10231) | |
|---|---|---|---|---|---|---|
| | MIC | MBC | MIC | MBC | MIC | MFC |
| **HL** | ≥500 | ≥500 | ≥500 | ≥500 | 31.3 | 190.9 |
| (**1**) [Cu(L)CH$_3$COO] | 15.6 | 31.3 | ≥500 | ≥500 | ≥500 | ≥500 |
| (**2**) [{Cu(L)Cl}$_2$]·H$_2$O | 3.9 | 7.8 | 250.0 | 500 | 15.6 | 73.5 |
| (**3**) [Cu(L)H$_2$O·DMF]NO$_3$ | 3.9 | 7.8 | ≥500 | ≥500 | ≥500 | ≥500 |
| (**4**) [Cu(L)Br] | 3.9 | 3.9 | 62.5 | 120.6 | 31.3 | 66.5 |
| (**5**) [Cu(L)H$_2$O]ClO$_4$ | 3.9 | 7.8 | ≥500 | ≥500 | ≥500 | ≥500 |
| Nitrofurazone | 4.7 | 9.4 | 4.7 | 4.7 | - | - |
| Miconazole | - | - | - | - | 16.0 | 76.9 |

Note: MIC—minimum inhibitory concentration; MBC—minimum bactericidal concentration; MFC—minimum fungicidal concentration; «-»—data not available.

The data obtained highlight that copper complexes exhibit the highest level of antimicrobial activity. Among the tested complexes, the most significant activity is observed against *S. aureus*, surpassing the activity of the proligand **HL**. In the case of complexes **2–5**, their activity is even higher than that of nitrofurazone. While **HL** and complexes **1**, **3**, and **5** show no activity against *E. coli*, complexes **2** and **4** demonstrate moderate activity. Conversely, complexes **1**, **3**, and **5** do not display any activity against the fungal strain *C. albicans*. **HL** and complexes **4** exhibit moderate activity, whereas complex **2** demonstrates stronger activity than miconazole.

### 2.2.2. Antioxidant Activity

Free radicals play a significant role in various detrimental biological processes, including protein denaturation and lipid peroxidation, contributing to the development of numerous human diseases [39,40]. Hence, investigating the antioxidant potential of the synthesized compounds becomes crucial to determine whether they can mitigate the levels of free radicals and provide protection against oxidative stress in the human body. The antioxidant activity against the 2,2-azinobis-(3-ethylbenzothiazoline-6-sulphonate) radical cation (ABTS$^{\bullet+}$) was evaluated for the compounds under examination: **HL** and copper(II) complexes **1–5**. The obtained results, represented as semi-maximal inhibitory concentrations (IC$_{50}$), are presented in Table 5.

The ligand **HL** and complexes **2–5** exhibit remarkable antioxidant activity, surpassing the activity of Trolox, a standard antioxidant utilized in medical applications. The tested ligand **HL** demonstrates an activity that is four times higher than that of Trolox. Complexes showed antioxidant activity against ABTS$^{\bullet+}$ with IC$_{50}$ of 10.1–47.4 $\mu$M. Among the copper(II) complexes, the antioxidant potency follows this sequence: [Cu(L)H$_2$O]ClO$_4$ ≥ [Cu(L)H$_2$O·DMF]NO$_3$ ≥ [{Cu(L)Cl}$_2$]·H$_2$O ≥ [Cu(L)Br] ≥ [Cu(L)CH$_3$COO].

**Table 5.** Antioxidant activity of the tested compounds ligand **HL** and copper(II) complexes **1–5** against cation radicals ABTS$^{\bullet+}$.

| Compound | IC$_{50}$, µM |
|---|---|
| **HL** | $8.5 \pm 1.5$ |
| (**1**) [Cu(L)CH$_3$COO] | $47.4 \pm 1.9$ |
| (**2**) [{Cu(L)Cl}$_2$]·H$_2$O | $24.3 \pm 1.3$ |
| (**3**) [Cu(L)H$_2$O·DMF]NO$_3$ | $23.3 \pm 0.9$ |
| (**4**) [Cu(L)Br] | $32.4 \pm 1.6$ |
| (**5**) [Cu(L)H$_2$O]ClO$_4$ | $10.1 \pm 0.3$ |
| Trolox | $33.3 \pm 0.2$ |

2.2.3. Acute In Vivo Toxicity of the Tested Compounds Assessed through Immobilization of the Crustacean *Daphnia magna*

To determine the toxicity of the tested compounds, the immobilization test on the crustacean *Daphnia magna* was conducted following a European Standardized Methodology. International organizations for animal protection recommend conducting in vivo toxicity research on *Daphnia magna*. In this context, as an alternative method used in this study, the complete replacement of animal toxicity testing with tests on invertebrate organisms was employed. This paper includes relevant information regarding the results of an experiment aimed at assessing the toxicity of the tested compounds through acute toxicity bioassays on an aquatic organism species from the arthropod subphylum, such as the crustaceans represented by *D. magna*. This organism is frequently used in laboratory experiments due to its structure, transparency, and ability to survive under a coverslip, making it easily observable under a microscope [40].

The test allowed for the evaluation of the acute toxicity of the tested compounds on *D. magna* at 24 h, expressed as the median lethal concentration (LC$_{50}$), which was calculated using the dose–response relationship determined by the least squares fitting method with the assistance of GraphPad software. All data are presented as means $\pm$ standard deviation (SD). Thus, the LC$_{50}$ values were determined, and the assessment of the effects on aquatic organisms was conducted. The LC$_{50}$ of the tested compounds were used as quantitative indicators of their toxicity and for the comparative evaluation of the obtained results.

Microscopic analysis of the control *D. magna* organisms that were not exposed to chemical compounds did not reveal any pathological changes. The effect of the compounds at the median lethal concentration on *D. magna* was determined through microscopic examination, indicating slight movements in over 50% of these invertebrate organisms. Additionally, it was observed that a significant portion of the *D. magna* remained immobile, especially at high concentrations of the chemical compound, as they exhibited a total cytotoxic effect. Upon examination, it was noted that the limbs and bodies of *D. magna* were deformed, and their contents were mixed with the growth media (Table 6).

**Table 6.** Results of the *Daphnia magna* bioassay for toxicity indicator determination of the tested compounds ligand **HL** and copper(II) complexes **1–5**.

| Compound | LC$_{50}$ (µM) |
|---|---|
| **HL** | $\geq 100$ |
| (**1**) [Cu(L)CH$_3$COO] | $3.5 \pm 2.8$ |
| (**2**) [{Cu(L)Cl}$_2$]·H$_2$O | $\geq 100$ |
| (**3**) [Cu(L)H$_2$O·DMF]NO$_3$ | $8.9 \pm 1.3$ |
| (**4**) [Cu(L)Br] | $1.0 \pm 0.1$ |
| (**5**) [Cu(L)H$_2$O]ClO$_4$ | $65.4 \pm 11.8$ |

As shown in Table 6, **HL** and the complex **2** dimers with LC$_{50} \geq 100$ µM have practically no impact on *D. magna*, whereas complexes **1, 3–5** exhibit toxicity with LC$_{50}$ values ranging from 1.0 to 65.4 µM after 24 h of exposure.

## 3. Materials and Methods

All the reagents used were chemically pure 3*d* metal salts $Cu(CH_3COO)_2 \cdot H_2O$, $CuCl_2 \cdot 2H_2O$, $CuBr_2$, $Cu(NO_3)_2 \cdot 3H_2O$, $Cu(ClO_4)_2 \cdot 6H_2O$ (Merck, Darmstadt, Germany) were used as supplied 2-Acetylpyridine was used as received (Sigma-Aldrich, Munich, Germany). The solvents were purified and dried according to standard procedures [41].

### 3.1. Synthesis

3.1.1. Synthesis of *N*-[4-({2-[1-(pyridin-2-yl)ethylidene]hydrazinecarbothioyl}amino)-phenyl]acetamide (HL)

The mixture consisting of 0.192 g (1 mmol) *N*-(4-isothiocyanatophenyl)acetamide, 0.135 g (1 mmol) 2-[1-hydrazinylideneethyl]pyridine, and 5 mL of THF is stirred at 50 °C for 3 h. It is distilled the solvent, then the solid is recrystallized from ethanol. The following are obtained: 0.301 g (92%).

**HL** (aciform pale yellow crystals), m.p. = 188–190 °C; $R_f$ = 0.59 (ethyl acetate-benzene 2:1). Elemental analysis for $C_{16}H_{17}N_5OS$, calc. (%): C, 58.7; N, 21.4; Found, (%): C, 58.6; N, 21.3.

$^1$H-NMR (DMSO-d$_6$) δ (ppm), 400 MHz: 2.05, s, 3H(CH$_3$CO); 2.51, s, 3H (CH$_3$-C=N); 7.41; 7.42; 7.56; 7.58, m, 4H (phenyl); 8.52; 8.54; 8.59; 8.60, m, 4H (pyridine); 10.00, s, 1H {HNC(S)}; 10.14, s, 1H {HNC(O)}; 10.60, s, 1H (NH-N=) the full spectrum in Figure S1.

$^{13}$C-NMR (DMSO-d$_6$) δ (ppm) 100 MHz: 12.9, (CH$_3$-C=N); 24.4, (CH$_3$CO); 119.1; 126.1; 132.6; 134.5; 137.4; 144.8 (phenyl); 148.9, (azomethine) 121.7; 127.1; 136.9; 149.5; 154.9, (pyridine); 168.7, (C=O); 177.8, (C=S) the full spectrum is shown in Figure S2.

Selected FT-IR data, ν (cm$^{-1}$): 3252 (N-H, amide II); 3196(N4-H); 3043(C-H, aryl/py); 2996(C-H, CH$_3$ from py); 2967 (C-H of CH$_3$ amide II); 1657 (C=O, amide II); 1611 (N-H); 1578 (C=N, azomethine); 1513 (C=C, aryl); 1486 (C=C, aryl); 1463, 1403(C=C, from py); 1366(C-N); 1299 (C=S) stretching; 1041(N-N); 841(C=S); 830 (1.4-sub,); 620 (py in plan), the full spectrum is shown in Figure S3.

3.1.2. Synthesis of Copper(II) Complexes (**1–5**)

[Cu(L)CH$_3$COO] (**1**)

Copper(II) acetate $Cu(CH_3COO)_2 \cdot H_2O$ (0.1996 g, 1 mmol) was added to a hot (50–55 °C) ethanolic solution (10 mL) of *N*-[4-({2-[1-(pyridin-2-yl)ethylidene]hydrazinecarbothioyl}amino)phenyl]acetamide HL (0.3274 g, 1 mmol). The mixture was stirred for 1 h under reflux. By cooling to room temperature, the green precipitate was obtained. It was filtered out, washed with cold ethanol, and dried in vacuo. Green solid. Yield: 92%; m.p. = >300 °C; FW: 448.98 g/mol; Anal. Calc. for $C_{18}H_{19}CuN_5O_3S$: C, 48.15; H, 4.27; Cu, 14.15; N, 15.60; S, 7.14; Found: C, 48.21; H, 4.46; Cu, 14.01; N, 15.46; S, 7.39. Selected FT-IR data, ν (cm$^{-1}$): ν(NH amide II) 3154, ν(CH Aryl) 3060, ν(C=Npy) 1509, ν(C–S) 675, the full spectrum is shown in Figure S4. Molar electrical conductivity (EtOH): 36 μS/cm.

[{Cu(L)Cl}$_2$]·H$_2$O (**2**)

The coordination compound **2** was synthesized similarly to compound **1** using $CuCl_2 \cdot H_2O$ (0.1705 g; 1 mmol) and HL (0.3274 g; 1 mmol). Brown solid. Yield: 89%; m.p.= >300 °C; FW: 868.81 g/mol; Anal. Calc. for $C_{32}H_{32}Cl_2Cu_2N_{10}O_2S_2$: C, 45.17; H, 3.79; Cu, 14.94; N, 16.46; S, 7.54; Found: C, 45.33; H, 3.81; Cu, 14.82; N, 16.52; S, 7.37. Selected FT-IR data, ν (cm$^{-1}$): ν(NH) 3375–3214, ν(CH Aryl) 3070, ν(C=Npy) 1556, ν(C–S) 670, the full spectrum is shown in Figure S5. Molar electrical conductivity (EtOH): 42 μS/cm.

[Cu(L)H$_2$O·DMF]NO$_3$ (**3**)

The coordination compound **3** was synthesized similarly to compound **1** using $Cu(NO_3)_2 \cdot 3H_2O$ (0.2416 g; 1 mmol) and HL (0.3274 g; 1 mmol). Green solid. Yield: 81%; m.p.= >300 °C; FW: 543.06 g/mol; Anal. Calc. for $C_{19}H_{25}CuN_7O_6S$: C, 42.02; H, 4.64; Cu, 11.70; N, 18.05; S, 5.90; Found: C, 42.20; H, 4.51; Cu, 11.86; N, 18.16; S, 5.98. Selected

FT-IR data, ν (cm$^{-1}$): ν(NH) 3268, ν(CH Aryl) 3062, ν(C=Npy) 1508, ν(C–S) 689, the full spectrum is shown in Figure S6. Molar electrical conductivity (EtOH): 102 μS/cm.

### [Cu(L)Br] (**4**)

The coordination compound **4** was synthesized similarly to compound **1** using copper(II) bromide CuBr$_2$ (0.2234 g, 1 mmol) and HL (0.3274 g; 1 mmol). Green solid. Yield: 92%; m.p.= >300 °C; FW: 469.84 g/mol; Anal. Calc. for C$_{16}$H$_{16}$BrCuN$_5$OS: C, 40.90; H, 3.43; Cu, 13.52; N, 14.91; S, 6.82; Found: C, 40.41; H, 3.57; Cu, 13.49; N, 14.82; S, 6.78. Selected FT-IR data, ν (cm$^{-1}$): ν(NH) 3318, ν(C=O) 1667, ν(C=N py) 1555, ν(C–S) 671, the full spectrum is shown in Figure S7. Molar electrical conductivity (EtOH): 47 μS/cm.

### [Cu(L)H$_2$O]ClO$_4$ (**5**)

The coordination compound **5** was synthesized similarly to compound **1** using Cu(ClO$_4$)$_2$·6H$_2$O (0.3705 g; 1 mmol) and HL (0.3274 g; 1 mmol). Green solid. Yield: 78%; m.p. = >300 °C; FW: 507.41 g/mol; Anal. Calc. for C$_{16}$H$_{18}$ClCuN$_5$O$_6$S: C, 37.87; H, 3.58; Cu, 12.52; N, 13.80; S, 6.32; Found: C, 37.98; H, 3.36; Cu, 12.68; N, 13.66; S, 6.21. Main FT-IR peaks (cm$^{-1}$):ν(NH) 3331, ν(CH Aryl) 3071, ν(C=Npy) 1567, ν(C–S) 673, the full spectrum is shown in Figure S8. Molar electrical conductivity (EtOH): 106 μS/cm.

### *3.2. FT-IR Spectroscopy*

FTIR spectra were recorded at room temperature using the BRUKER ALPHA spectrometer, in the wavelength range 4000–400 cm$^{-1}$, in the scientific research laboratory "Advanced Materials in Biopharmaceutics and Technics" of the State University of Moldova, Republic of Moldova. The spectral results were interpreted using the OPUS version 7.5 program.

### *3.3. NMR Spectroscopy*

Nuclear Magnetic Resonance (NMR) $^1$H, $^{13}$C, NMR spectra were recorded at room temperature using the BRUKER DRX-400 spectrometer (Institute of Chemistry, State University of Moldova, Republic of Moldova). Chemical shifts are measured in ppm relative to tetramethylsilane (TMS), as solvents were used: DMSO-d$_6$. The obtained results were processed using the MestReNova v 14.1.2 program.

### *3.4. Molar Conductivity*

Most dissolved substances in water/protic organic solvents dissociate into ions that conduct electricity. Conductometric analysis was performed on the ADWA AD8000 (pH/mV/EC/TDS and Temperature Meter). Calibration of the electrode (AD 76309) was performed with standard solutions of 1430.0 μS/cm and 12,880.0 μS/cm. Samples were solubilized in H$_2$O/DMF/DMSO/EtOH or mixtures of the listed solvents. The concentration of the investigated samples was $1 \cdot 10^{-3}$M [8,42].

### *3.5. Melting Point*

The substance sample, dry and finely pulverized beforehand by drying the crystals on a watch glass, is introduced into a capillary with a diameter of about 1 mm welded at one end. The height of the substance layer in the capillary should be 4–6 mm. The substance is introduced into the capillary by repeated "knocks" on a hard surface. Recorded the melting point on the Stuart$^{®}$ SMP10 Apparatus, in the range of ambient temperature to 300 °C with a resolution of 1 °C.

### *3.6. Thin Layer Chromatographic*

Thin-layer chromatography, also called partition chromatography, is based on the differences between the partition coefficients of the substances being separated. The analysis was performed using chromatographic plates (Macherey-Nagel, 0.2 mm Silica gel 60 with fluorescent indicator UV254) [43].

### 3.7. X-ray Crystallography

Single-crystal X-ray diffraction measurements for ligand **HL** and coordination compounds **2–4** were carried out on an XCalibur E charge-coupled device (CCD) diffractometer equipped with a CCD area detector and a graphite monochromator utilizing MoK$\alpha$ radiation at room temperature. Final unit cell dimensions were obtained and refined on an entire data set. All calculations necessary to solve the structures and to refine the proposed model were carried out with the SHELXS97 and SHELXL2015 program packages [44,45]. The nonhydrogen atoms were treated anisotropically (full-matrix least squares method on F2). The H atoms were placed in calculated positions and were treated using riding model approximations with Uiso(H) = 1.2Ueq(C) and Uiso(H) = 1.5Ueq(O). The disordered water molecules were found in compound **HL**. The X-ray data and the details of the refinement of studied compounds are summarized in Table 1, the selected bond lengths and angles as well as hydrogen bond parameters are given in Tables 2 and 3. The geometric parameters were calculated by the PLATON program and Mercury software was used for visualization of structures. The hydrogen atoms that were not involved in the hydrogen bonding were omitted from the generation of the packing diagrams.

### 3.8. EPR Study

EPR data were acquired using a Bruker Elexsys E 500 spectrometer, operating at a microwave frequency of approximately 9.47 GHz. Spectra were recorded with a microwave power of 10 mW over a sweep width of 200 mT, centered at 320 mT, and a modulation amplitude of 0.4 mT. These experiments were conducted at a temperature of 110 K using a liquid nitrogen cryostat.

To prepare the samples, approximately 10 μmol of the respective compound was dissolved in 1 mL of DMSO, resulting in a 10 mM stock solution. Solutions of 0.1 mM, 0.2 mM, and 0.5 mM were prepared by diluting the stock solution with DMSO. For low-concentration measurements, multiple scans (4 to 6) were averaged [46].

### 3.9. Antibacterial and Antifungal Activity

Standard strains of *Staphylococcus aureus* (ATCC 25923), *Escherichia coli* (ATCC 25922), and *Candida albicans* (ATCC 10231) were used to determine antibacterial and antifungal activities of the proligand **HL** and copper(II) coordination compounds **1–5**. The antibacterial and antifungal activity of the synthesized compounds was evaluated using the microdilution broth test, which allowed us to determine the minimum inhibitory concentration (MIC), minimum bactericidal concentration (MBC), and minimum fungicidal concentration (MFC). We followed established reference methods: the Third Edition, 2002 of the "Reference Method for Broth Dilution Antifungal Susceptibility Testing of Yeasts" (Clinical and Laboratory Standards Institute document M27-A3) for fungi and the 9th edition, 2012 of the "Methods for Dilution Antimicrobial Susceptibility Tests for Bacteria that Grow Aerobically" (Clinical and Laboratory Standards Institute) for bacteria.

For MIC assays, a stock solution of each tested compound (10 mg mL$^{-1}$) was prepared in dimethyl sulfoxide (DMSO). This stock solution was then diluted in Muller Hinton Broth (MHB) for bacteria and liquid RPMI (Roswell Park Memorial Institute) 1640 medium with L-glutamine and 0.165 M MOPS buffer (without sodium bicarbonate) for fungi. Subsequent dilutions were made using 2% of peptonate bullion. Plates were covered and incubated on a shaker at 37 °C for 24 h (bacteria), and 48 h (*Candida* spp.). MICs were visually assessed after the respective incubation period, and the lowest sample concentration with no (or minimal) growth was recorded.

To determine the minimum bactericidal concentrations (MBC), 10 μL aliquots from wells with no microorganism growth were plated on Mueller–Hinton Agar (for bacteria) or Sabouraud Dextrose Agar (for fungi) and incubated at 37 °C for 24 h (bacteria), 48 h (*Candida* spp.). The lowest concentration that resulted in no growth after subculturing was considered the MBC or MFC. *Furacillinum* served as the standard antibacterial drug,

while nystatin was used as the standard antifungal drug. All experiments were conducted in triplicate.

### 3.10. Antioxidant Activity

The antioxidant activity of the synthesized compounds was assessed using the ABTS$^{.+}$ method as described by Re et al. [47] with some modifications. The ABTS$^{.+}$ radical cations were generated by mixing a 7 mM solution of ABTS (2,20-azino-bis(3-ethylbenzothiazoline-6-sulphonic acid)) (Sigma-Aldrich, USA) with a 2.45 mM solution of potassium persulfate ($K_2S_2O_8$) (Sigma-Aldrich, USA) at 25 °C in the dark for 12–20 h. The resulting solution was then further diluted with acetate-buffered saline (0.02 M, pH 6.5) to achieve an absorbance of $0.7 \pm 0.1$ at 734 nm.

To prepare the samples for testing, the synthesized compounds were dissolved in DMSO to create solutions with concentrations of 1, 10, and 100 µM. Subsequently, 20 µL of each solution was transferred to a 96-well microtiter plate, followed by the addition of 180 µL of the ABTS$^{.+}$ working solution, which was then thoroughly mixed. The decrease in absorbance at 734 nm was measured precisely after a 30 min incubation at 25 °C. All measurements were performed in triplicate, with DMSO serving as the negative control. Blank samples were also run using solvent without ABTS$^{.+}$.

The percent inhibition (I, %) of free radical cations ABTS·+ was calculated using the following Formula (1):

$$I(\%) = \frac{\text{Abs(control)} - \text{Abs(sample)}}{\text{Abs(control)}} * 100\% \tag{1}$$

where Abs734 nm (control) represents the absorbance of the control solution, and Abs734 nm (sample) denotes the absorbance in the presence of sample solutions or standards for positive controls. The $IC_{50}$ values were determined using the Hill equation.

### 3.11. Toxicity

The toxicity assessment of the tested compounds was conducted using *Daphnia magna* (Straus, 1820). The *Daphnia magna* used in this study were obtained from a parthenogenetic culture [12,47–50]. The experimental design adhered to ISO 6341: 2012 guidelines.

*Daphnia magna* were nourished with Chlorella vulgaris, a unicellular algae cultivated using aseptic techniques to prevent contamination by bacteria, algae, or protozoa. Chlorella vulgaris was grown in Prat's growth medium, which consisted of $KNO_3$ (1 µM), $MgSO_4 \cdot 7H_2O$ (40 µM), $K_2HPO_4 \cdot 3H_2O$ (400 µM), and $FeCl_3 \cdot 6H_2O$ (3.6 µM) in distilled water (pH adjusted to 7.0, autoclaved, and stored at 5 °C).

*D. magna* were maintained in aerated aqueous straw infusion growth media supplemented with $CaCl_2$ (11.76 g/L), $NaHCO_3$ (2.59 g/L), KCl (0.23 g/L), and $MgSO_4 \cdot 7H_2O$ (4.93 g/L) to maintain a pH of approximately $7.5 \pm 0.2$ and ensure dissolved oxygen levels of $\geq 6.0$ mg/L.

Juveniles were selected based on size and acclimated to fresh medium for 24 h. The *D. magna* were cultured in Costar® 24-well clear sterile multiple well plates, covered with lids to prevent contamination and evaporation while allowing gaseous exchange. Each well contained 10 daphnids in 1000 µL of each dilution of the tested compounds.

The bioassay was conducted with concentrations ranging from 0.1 to 100 µM (0.1, 1, 10, and 100 µM) to determine the $LC_{50}$ for each compound. Stock solutions were diluted with aqueous straw infusion growth media to achieve the required concentrations. The final test solutions contained up to 0.1% DMSO and had a final volume of 1 mL. A 0.1% DMSO solution in aerated medium (pH~$7.5 \pm 0.2$; $O_2 \geq 6.0$ mg/L) served as the negative control.

Throughout the experiment, juvenile daphnids were incubated at $22 \pm 2$ °C under a 16 h light/dark cycle (500–1000 lx). Mobility (viability) of the test organisms was assessed after the 24 h exposure. The experiment was conducted in triplicate.

Daphnids were considered immobilized if they did not swim during the 15 s period following gentle agitation of the test and control solutions, even if they could still move their antennae. The percentage of viability (V, %) of *Daphnia magna* was calculated using the Formula (2):

$$V(\%) = \frac{N(sample)}{N(control)} * 100\% \tag{2}$$

where N represents the number of viable *Daphnia magna*. $LC_{50}$ values, which represent the median lethal concentration that kills 50% of the juvenile daphnids, were determined using the least squares fit method based on the dose-response equation.

## 4. Conclusions

The new thiosemicarbazone skeleton functionalized with a medicinal fragment such as paracetamol was used as a ligand, resulting in the formation of the thiosemicarbazone **HL**. Five coordination compounds based on copper(II) salts were synthesized. **HL** coordinates with the central ion via the azomethine nitrogen atom, the pyridinic nitrogen, and the thionic sulfur atom. Most of the coordination compounds (**1, 3–5**) are exclusively monomers, while compound **2** forms a dimer through the sulfur atom of the adjacent molecule.

The structures of **HL** and complexes **1–3** have been determined using single-crystal X-ray diffraction analysis. The **HL** ligand is in a non-deprotonated form, and it is deprotonated in the case of compounds **1–3**. The copper atom in **1–3** is five-coordinated in a distorted square–pyramidal coordination geometry. In the crystal, these compounds form centrosymmetric dimers where the monomers are held together by a bridge sulfur atom (in complex **2**) and hydrogen bonds.

Two mononuclear Cu(II) species **1, 3–5** can be observed in solution of DMSO by EPR studies.

All compounds were tested for antimicrobial, antifungal, and antioxidant activities, and their toxicity to *Daphnia magna* was studied. Biological evaluation has revealed that most of the synthesized compounds demonstrate promising antibacterial, antifungal, and antiradical activities. In many cases, their antibacterial/antifungal activity is comparable to that of certain drugs used in medicine for these purposes, and in some cases, even surpasses them. **HL** and complexes **2–5** exhibit antioxidant activity that surpasses that of Trolox which is used in medical practice. Furthermore, **HL** and complexes **2**, and **5** display virtually no toxicity to *D. magna*.

**Supplementary Materials:** The following supporting information can be downloaded at: https://www.mdpi.com/article/10.3390/inorganics11100408/s1, Figure S1. $^1$H-NMR spectrum of thiosemicarbazone **HL**; Figure S2. $^{13}$C-NMR spectrum of thiosemicarbazone **HL**; Figure S3. FT-IR spectrum of **HL**; Figure S4. FT-IR spectrum of the coordination compound [Cu(L)CH$_3$COO] (**1**); Figure S5. FT-IR spectrum of the coordination compound ([Cu(L)Cl]$_2$·H$_2$O (**2**); Figure S6. FT-IR spectrum of the coordination compound [Cu(L)(H$_2$O)(DMF)]NO$_3$ (**3**); Figure S7. FT-IR spectrum of the coordination compound [Cu(L)Br] (**4**); Figure S8. FT-IR spectrum of the coordination compound [Cu(L)(H$_2$O)]ClO$_4$ (**5**); Figure S9. RES spectrum of the coordination compound (**1–5**).

**Author Contributions:** Conceptualization, A.G.; methodology, R.R. and O.G.; validation, R.R., O.G., V.T. and A.G.; formal analysis, O.G. and R.R.; investigation, O.G., C.H. and D.I.; resources, C.H., D.I. and A.G.; data curation, A.G.; writing—original draft preparation, V.T. and Y.C.; writing—review and editing, R.R., O.G. and A.G.; visualization, R.R., C.H. and Y.C.; supervision, A.G.; project administration, A.G. All authors have read and agreed to the published version of the manuscript.

**Funding:** This work was fulfilled with the financial support of the National Agency for Research and Development. Projects 20.80009.5007.10; 20.80009.7007.12 and 20.80009.5007.15.

**Data Availability Statement:** Data are contained within the article or supplementary material.

**Acknowledgments:** The authors thank Greta Balan and Olga Burduniuc for their help in performing antibacterial and antifungal testing.

**Conflicts of Interest:** The authors declare no conflict of interest.

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
