# Peer review of "Synthesis, Characterization, and Biological Properties of the Copper(II) Complexes with Novel Ligand: N-[4-({2-[1-(pyridin-2-yl)ethylidene]hydrazinecarbothioyl}amino)phenyl]acetamide"

_inorganics, doi:10.3390/inorganics11100408_

Round 1

Reviewer 1 Report

The manuscript ” Synthesis, characterization, and biological properties of the Copper(II) Complexes with novel ligand: N-[4-({2-[1-(pyridin-2-yl)ethylidene]hydrazinecarbothioyl}amino)phenyl]acetamide” by authors: Roman Rusnac, Olga Garbuz, Yurii Chumakov, Victor Tsapcov, Christelle Hureau, Dorin Istrati and Aurelian Gulea, follows the process of synthetic route optimization, as well as a full chemical characterization of paracetamol based thiosemicarbazone molecule and theirs neutral and cationic copper(II) complexes, as well as antibacterial, antifungal and antioxidant activities. The manuscript is well written and all presented characterization data correspond to the proposed structures. Only a few minor typo corrections should be done:

Page 3, line 85 authors should write Figure S1 and S2, not only S2, since the presence of tautomers could be seen in both of NMR spectra (1H and 13C).

Page 8, line 147 authors should erase S34, it should be only S3.

Page 8, line 144: I think that ligand coordination via two atoms is usually called bidentate not didentate.

Page 14, line 415: in the formula MgSO4, 7H2O authors should erase the comma and write the correct formula. 

Taking all into account I recommend this manuscript for publication in Inorganics after minor typo corrections.

Author Response

Thank you very much for taking the time to review this manuscript. Please find the detailed responses below and the corresponding revisions/corrections highlighted/in track changes in the resubmitted files.

Comments 1: Page 3, line 85 authors should write Figure S1 and S2, not only S2, since the presence of tautomers could be seen in both of NMR spectra (1H and 13C).

Response 1: Thank you for pointing this out. We have added Figure S1 to the description of "Tautomeric forms of HL" on page 3, line 91.

Comments 2: Page 8, line 147 authors should erase S34, it should be only S3.

Response 2: Thank you for pointing this out. Revised accordingly. Page 8, line 156.

Comments 3: Page 8, line 144: I think that ligand coordination via two atoms is usually called bidentate not didentate.

Response 3: Thank you for pointing this out. Revised accordingly. Page 8, line 150.

Comments 4: Page 14, line 415: in the formula MgSO4, 7H2O authors should erase the comma and write the correct formula. 

Response 4: Thank you for pointing this out. We have corrected the formula to MgSO4·7H2O. Page 14, line 421.

Thank you very much for the positive review.

Reviewer 2 Report

The article can be published after some mentioned major revisions which can be addressed as follows:

1- Authors must be check the structure of HL new ligand by using mass spectroscopy.

2- Authors must be refer in the introduction section, why they synthesis of this type of ligand, it means the important of this ligand.

3-  Introduction section need to become more details that authors written with reduced informations.

4- Authors can make one mass spectra analysis for the synthesized complex.

5- The melting points of all complexes not recorded in the article?

6- Authors must be updated the references list to included 2022 and 2023 in the introduction section.

minor check

Author Response

Thank you very much for taking the time to review this manuscript. Please find the detailed responses below and the corresponding revisions/corrections highlighted/in track changes in the resubmitted files.

Comments 1: Authors must be check the structure of HL new ligand by using mass spectroscopy.

Response 1: Since we have the solid-state structure of this substance obtained through X-ray crystallography analysis and have conducted analyses in solution, including NMR and Fourier-transform infrared spectroscopy (FTIR), we believe that these data are sufficient for discussing the structure of the ligand HL. Unfortunately, in our laboratory in the Republic of Moldova, we do not have access to mass spectrometry due to the lack of the necessary equipment.

Comments 2: Authors must be refer in the introduction section, why they synthesis of this type of ligand, it means the important of this ligand.

Response 2: Thank you for your valuable suggestion. We have included a description in the introduction section explaining why we synthesized this type of ligand and emphasizing its importance.

Comments 3: Introduction section need to become more details that authors written with reduced informations.

Response 3: We agree with this comment. The Introduction section has been revised to include more detailed information as suggested.

Comments 4: Authors can make one mass spectra analysis for the synthesized complex.

Response 4: Since we have the solid-state structure of this substance obtained through X-ray crystallography analysis and have conducted analyses in solution, including NMR and Fourier-transform infrared spectroscopy (FTIR), we believe that these data are sufficient for discussing the structure of the synthesized complexes 1-5. Unfortunately, obtaining mass spectra of the complexes is not possible in the Republic of Moldova due to the lack of the necessary equipment.

Comments 5: The melting points of all complexes not recorded in the article?

Response 5: Thank you for bringing this to our attention. We have added the melting points of all complexes to 3.1.2. Synthesis of copper(II) complexes (1-5).

Comments 6: Authors must be updated the references list to included 2022 and 2023 in the introduction section.

Response 6: Thank you for pointing this out. The references list in the introduction section has been updated to include sources from 2022 and 2023.

Comments on the Quality of English Language -   minor check

Response: The English language has been edited for minor checks.

Thank you for your comments.

Round 2

Reviewer 2 Report

Accepted

Minor Check